# The Binding of CSL Proteins to Either Co-Activators or Co-Repressors Protects from Proteasomal Degradation Induced by MAPK-Dependent Phosphorylation

**DOI:** 10.3390/ijms232012336

**Published:** 2022-10-15

**Authors:** Johannes Fechner, Manuela Ketelhut, Dieter Maier, Anette Preiss, Anja C. Nagel

**Affiliations:** Department of General Genetics 190g, University of Hohenheim, Garbenstr. 30, 70599 Stuttgart, Germany

**Keywords:** activator complex, CSL, degron, MAPK, Notch signalling dynamics, suppressor of hairless, proteasomal degradation, protein stability, repressor complex

## Abstract

The primary role of Notch is to specify cellular identities, whereby the cells respond to amazingly small changes in Notch signalling activity. Hence, dosage of Notch components is crucial to regulation. Central to Notch signal transduction are CSL proteins: together with respective cofactors, they mediate the activation or the silencing of Notch target genes. CSL proteins are extremely similar amongst species regarding sequence and structure. We noticed that the fly homologue suppressor of hairless (Su(H)) is stabilised in transcription complexes. Using specific transgenic fly lines and HeLa RBPJ^KO^ cells we provide evidence that Su(H) is subjected to proteasomal degradation with a half-life of about two hours if not protected by binding to co-repressor hairless or co-activator Notch. Moreover, Su(H) stability is controlled by MAPK-dependent phosphorylation, matching earlier data for RBPJ in human cells. The homologous murine and human RBPJ proteins, however, are largely resistant to degradation in our system. Mutating presumptive protein contact sites, however, sensitised RBPJ for proteolysis. Overall, our data highlight the similarities in the regulation of CSL protein stability across species and imply that turnover of CSL proteins may be a conserved means of regulating Notch signalling output directly at the level of transcription.

## 1. Introduction

The Notch signalling pathway belongs to a handful of highly conserved signalling pathways that govern cellular differentiation and homeostasis throughout development in eumetazoans [1,2]. In contrast to other pathways acting at a longer range via secreted morphogens or ligands, Notch signalling is restricted to intercellular contacts, thereby allowing for a direct cell–cell communication [2,3]. The Notch receptor as well as all known Notch ligands are membrane tethered proteins, restricting ligand–receptor interactions to neighbouring cells or even within the same cell [2,3,4,5]. In accordance with a pivotal role in accurate cellular differentiation, Notch dysregulation has been associated with several congenital human diseases (reviewed in: [6,7,8]). Moreover, Notch has a central role in oncogenesis that has been extensively studied. In fact, Notch signalling can be both oncogenic but also tumour suppressive, strongly depending on cell type and context [7,8,9]. Clearly, Notch signalling activity has been associated with cellular proliferation in a cell-autonomous as well as a non-cell-autonomous fashion [1,2,4,7,8,10].

Cells are exquisitely sensitive in their response to the dose of Notch activity [3,8,11]. Accordingly, the availability and turn-over of ligands as well as of the Notch receptor itself are highly critical steps throughout the signalling event. As a consequence, these processes have been well scrutinised in the past (reviewed in: [2,3,4,5,12]). Notably, internalisation, degradation and turnover of the Notch receptor are very important steps in the regulation of signalling output, and have been linked to a number of diseases based on the malfunction of Notch activity (reviewed in: [7,8,11,13]).

The mechanism of Notch signal transduction is straightforward, as there is no amplification step by second messengers or by phosphorylation cascades involved. Instead, the Notch receptor is cleaved within the membrane upon ligand binding, releasing the biologically active Notch intracellular domain (NICD, also abbreviated ICN for intra-cellular Notch) [1,2,3,4,5]. Then, NICD itself becomes part of a transcriptional activator complex to induce Notch target gene expression [5,14]. In the heart of the activator complex lies the highly conserved CSL protein (acronym for human CBF1, *C-promoter Binding Factor 1*, corresponding to mammalian RBPJ, *Recombination signal Binding Protein*, for *D. melanogaster*
Su(H), *Suppressor of Hairless*, and for *C. elegans* Lag1, *lin-12 and glp-1*). CSL is a DNA-binding protein that recruits NICD and the co-activator *Mastermind* (MAM) to Notch target gene promoters [5,14]. The crystal structures of CSL bound to the DNA as well as within the activator complex demonstrate high conservation between vertebrates and invertebrates [14]. The primary contact by NICD is made with the so-called RAM domain (*RBPJ-Associated Molecule*) at the beta-trefoil domain of CSL, allowing tight binding together with the Notch ankyrin (ANK) repeats that then contact the C-terminal domain of CSL [2,14]. Both, NICD and MAM contain trans-activating domains themselves allowing transcription initiation [2,3,13,14,15]. However, work in the recent years has uncovered a battery of familiar and novel factors involved in reshaping the chromatin landscape to allow for a timely and appropriate transcriptional response after Notch receptor activation (reviewed in: [4,11,13,14,15,16]). Clearly, Notch signalling is exquisitely sensitive to the chromatin landscape. On the one hand, Notch activation resets the chromatin architecture close to Notch target gene promoters. On the other hand, chromatin modifications and histone rearrangements influence target gene specificity of Notch transcription complexes, and hence the Notch signalling output (reviewed in: [4,11,13,14,15,16]).

CSL proteins serve as molecular switches, mediating the silencing of Notch target genes in the absence of signal as well [17,18,19]. In this case, CSL assembles repressor complexes containing one or more co-repressors. In mammals, a large number of different co-repressors have been identified over the years in various cell lines and tissues (reviewed in: [13,16,19]). Interestingly, the majority of those factors studied in molecular detail bind the beta-trefoil domain of CSL in a manner similar to the Notch RAM domain. Consequently, these factors are thought to compete with NICD for the binding to CSL in a quantitative manner [16,20,21]. SHARP (*SMRT/HDAC1-Associated Repressor Protein*, corresponding to murine MINT, *Msx2-Interacting Nuclear Target*), however, binds to the C-terminal domain of CSL as well, increasing the binding affinity within the protein complex [19,22]. This interaction resembles the binding of the major Notch antagonist in *Drosophila* named Hairless (H). H binds exclusively to CSL’s C-terminal domain, thereby distorting its structure and precluding the binding of NICD [23,24]. Even though binding affinities of some of the co-repressors are within nanomolar range, NICD is able to displace them from repressor complexes [16,20,21,22,23,25]. Yet, it appears unlikely that displacement takes place at the DNA in vivo, i.e., that repressor complexes are converted to activator complexes by an exchange of CSL’s cofactors. Firstly, albeit the DNA-binding site of CSL is extremely well conserved between vertebrates and invertebrates, DNA-binding occurs with only modest affinity, allowing for a rapid exchange of CSL at the Notch target gene promoters [26,27]. Secondly, CSL proteins rely on their cofactors for nuclear entry in mammals and in *Drosophila* alike, indicating that activator and repressor complexes assemble within the cytoplasm and not within the nucleus [17,28,29,30]. Moreover, occupancy of the Notch target gene promoters by Su(H) or RBPJ is rather transient and only stabilised upon the activation of Notch [31,32,33,34,35,36]. Overall, DNA-binding of CSL transcription complexes appears rather dynamic. Yet, minimal concentrations of CSL–NICD activation complexes suffice for Notch signalling to take place. Perhaps the cofactors recruited by CSL generate a microenvironment in which the components of the transcription machinery cluster at a higher local concentration, thereby resulting in high transcription bursts [11].

The dose sensitivity of Notch signalling, on the one hand, and the central role of CSL proteins in signal transduction, on the other hand, raise the possibility for a regulation of Notch activity via CSL availability. Limited availability of CSL would open the possibility for a tight control of Notch signalling output at the level of transcription. Since most of mammalian co-repressors require more energy for the binding of CSL than NICD, restriction of CSL levels would preferentially affect the formation of repressor complexes over that of activator complexes [37]. In fact, CSL availability is regulated by several external and internal cues. For example, selective degradation of human CBF1/RBPJ, resulting in an inhibition of Notch signalling activity is mediated by VEGF (vascular endothelial growth factor) in support of angiogenic sprouting [38]. Moreover, p38 MAPK regulated the proteolysis of CBF1/RBPJ in HEK293 cells by a Presenilin2-dependent phosphorylation [39]. More recently, a link between RBPJ turnover and tumorigenesis was uncovered. In this report it was shown that metabolic stress resulted in a cyclin F-dependent polyubiquitylation of RBPJ in glioma cells, followed by proteasomal degradation. Moreover, cyclin F levels correlated with glioma formation in a mouse tumour model [40]. Despite the underlying principle of regulating Notch activity via RBPJ proteolysis, reports on the half-life of RBPJ vary dramatically between two and twelve hours [38,39,40].

The Drosophila orthologue Su(H) was likewise predicted to be a rather stable protein based on earlier observations [41,42,43]. However, studies using H-binding defective Su(H) mutants indicate that Su(H) is only stable within protein complexes [44]. In this work, specific Su(H) mutants were generated by genome engineering at the endogenous locus and protein expression was analysed in vivo. The conservative exchange of three leucine residues to alanine, known to contact H from the crystal structure of the Su(H)–H repressor complex, completely abolished the binding to H [24,44]. The work uncovered that Su(H)^LLL^ protein was largely unstable and localised primarily to the cytosolic compartment, in agreement with the twofold role of H in nuclear translocation and stabilisation of Su(H) protein [44]. Thus, unbound solitary Su(H) could be a relatively unstable protein whose apparent stability is due to the influence of its binding partners. In this case, availability of Su(H) could be under the influence of cofactors able to bind it directly or to promote complex formation. Such a scenario might explain the variance in the reported RBPJ half-life. Moreover, some of the many known CLS-binding partners might influence RBPJ stability, thereby substantially extending their regulatory potential [16,37]. To further our understanding on the mechanisms underlying Su(H) protein stability, we generated several constructs to be specifically induced in vivo in the fly or in a HeLa RBPJKO cell line model. Our data show that solitary Su(H) has a half-life of about two hours that is significantly extended by the binding to either the co-repressor H or the co-activator NICD. In contrast, both human and mouse RBPJ are largely resistant to degradation in HeLa cells. The latter becomes slightly more susceptible when mutating presumptive contact sites within the beta-trefoil domain, i.e., the RAM-binding domain. We provide evidence that Su(H) is subjected to proteasomal degradation, presumably ubiquitinated at multiple interchangeable sites. Finally, we show that Su(H) proteolysis is controlled by MAPK-dependent phosphorylation as well, highlighting the similarities in the regulation of CSL protein stability across species.

## 2. Results

### 2.1. H Binding-Defective Su(H)^LLL^ Protein Is Degraded More Rapidly Than Wildtype Su(H) Protein

Based on various observations within previous analyses, it was predicted, that Su(H) protein is a rather stable protein [41,42,43]. However, studies using H-binding defective Su(H) mutants indicate, that Su(H) is only stable within the activator or repressor complex [44]. Thus, unbound solitary Su(H) could be a relatively unstable protein whose apparent stability is due to the influence of its binding partners. To investigate the stability of the Su(H) protein, we generated transgenic fly lines expressing the wildtype Su(H) protein and the H-binding defective Su(H)^LLL^ protein under the control of the heat-shock promoter element [45,46,47,48]. This enables a transient overexpression of the Su(H) variants and thus the tracking of the degradation of these protein variants. We employed site-specific recombination to insert the various constructs at the identical chromosomal position (3R 96E) thereby avoiding position effects [23,49]. To distinguish between overexpressed and endogenous proteins, the Su(H) variants were fused to a myc tag. After a 30 min heat shock at 37 °C, immuno-histochemical staining of wing imaginal discs was performed at defined time points (0 h, 1 h, 2 h, 4 h, 8 h and 12 h) after heat shock. It was expected that the Su(H)^LLL^ protein variant would show a significantly faster degradation than the wild-type protein, based on the lack of H-binding. Initially, there is a similarly strong expression of the Su(H)^wt-myc^ and the Su(H)^LLL-myc^ variant, which reaches its maximum about 1 h after the heat shock. Two hours after the induction, a slight decrease in the abundance of either protein variant is already observed (Figure 1a,b). At four hours after the induction, protein levels dropped. At this time point, the accumulation of the Su(H)^LLL^^-myc^ protein was significantly reduced compared to the wildtype protein as determined by a quantification of the signal intensity (Figure 1b). This agrees with a greater instability of the Su(H)^LLL^^-myc^ variant, which may be initially stabilised by the presence of the activated Notch protein [44]. Whereas protein amounts were nearly constant for the next four hours, they further dropped 12 h post-induction, and in the case of the Su(H)^LLL^^-myc^ protein nearly reached the pre-induction level. Considering the percental decline, it is obvious that the Su(H)^wt^^-myc^ variant is remarkably more stable than the Su(H)^LLL^^-myc^ variant for which a half-life of approximately 2 h was determined (Figure 1c).

Since the two protein variants only differ in their ability to bind to H, it can be assumed that Su(H)^wt^^-myc^ was stabilised by the endogenous H protein. In the early phase, i.e., after the initial peak of induction at 1 h, Su(H)^wt^^-myc^ was subjected to degradation very similar to the Su(H)^LLL^^-myc^ protein but then appeared to be stabilised (Figure 1c). This could be explained by the fact that overexpressed Su(H) protein is in excess relative to its binding partners, and that this excessive protein is as rapidly degraded as is the Su(H)^LLL^ protein which cannot bind to the co-repressor H. In this case, the protein levels of the binding partner H determine Su(H) protein levels.

### 2.2. Su(H) Protein Is Subjected to Proteasomal Degradation

A major pathway of protein degradation involves the proteasome [50]. To address whether Su(H) is subjected to proteasomal degradation, we employed the elegant in vivo method first established by F. Schweisguth [51] (Figure 2a). Here, the protein under investigation is transiently and ubiquitously expressed in the background of the dominant temperature sensitive DTS5 mutant affecting the Pros β6 subunit of the proteasome [51,52]. Under restrictive conditions, i.e., temperatures above 29 °C, DTS5 prevents proteasome function, i.e., protein degradation, which is completely normal at lower temperatures [51]. To allow for a tissue-specific overexpression of DTS5, the Gal4-UAS system was employed: UAS-DTS5 combined with ptc-Gal4 was expressed along the antero–posterior boundary of imaginal discs [53,54]. After rearing the larvae at permissive temperature, they were shifted for a day to 29 °C to block proteasomal activity before applying the heat shock. In this background, accumulation of heat-induced Su(H)^wt-myc^ or Su(H)^LLL-myc^ protein was examined over time (Figure 2a and Appendix A). As outlined above, ectopic Su(H) protein is well visible for one to two hours and decreases conspicuously at the four-hour time point (Figure 1). Yet, both Su(H)^wt-myc^ and Su(H)^LLL-myc^ protein variants resisted the degradation along the antero–posterior boundary, i.e., in the domain of proteasomal dysfunction (Figure 2b,c and Appendix A), indicating that either protein isoform is subjected to proteasomal degradation. Notably, signals from the resistant proteins appeared lower at the eight-hour time point, suggesting the lysosomal pathway may add to Su(H) protein turnover [50].

Proteins destined for degradation by the proteasome are poly-ubiquitinated at specific lysine residues [55]. The Su(H) protein harbours five lysine residues predicted in silico with the UbPred software as likely ubiquitination sites [56]: K103, K104, K112, K389 and K393. The *PhosphositePlus* database reports ubiquitination of lysine 343 in murine Rbpj as secondary modification, corresponding to K389 in Su(H) [57,58]. Moreover, a high throughput analysis recently identified the same site in human RBPJ protein to be ubiquitinated in Jurkat and Hep2 cells as well [59]. Interestingly, the surrounding residues are highly conserved between mammals and insects, suggesting a likewise modification in Su(H) [60,61] (Appendix A). To study the contribution of this lysine residue to Su(H) protein turnover, K389 and the adjacent K393 were changed to alanine. Transgenic flies expressing the respective mutant under the control of the heat shock element were combined with ptc-Gal4 and UAS-DTS5 to study the stability of the induced proteins in the presence and absence of proteasomal activity as outlined above (Appendix A). We expected a stabilisation of the Su(H) protein if one or both lysine residues were essential targets for ubiquitination, and hence for recognition by the proteasome. In this case, we might have expected no, or a very slow, degradation of heat induced protein all over the tissue and not just within the ptc-DTS5 domain of proteasome dysfunction. This was, however, not the case, as the K389/393A mutant was largely indistinguishable from the wildtype (Appendix A). Likewise, the quintuple mutant Su(H)^5KA^ protein (K103/104/112/389/393A replacement) displayed a similar turnover (Appendix A). As E3 ligases may use other lysine residues in place of their preferred target, we decided to expand the number of lysine mutations. We included the surface-exposed lysine residues K200, K343 and K359 that are in the vicinity of K389 (Appendix A). In fact, the corresponding residues in human RBPJ are ubiquitinated in Hep2 and in Jurkat cells as well [59]. We generated a quintuple mutant Su(H)^5KR^ protein with lysine replacements to arginine (K200, 343, 359, 389, 393R) not to interfere with the overall charge of Su(H) to avoid any impact on structure and function [14]. Yet, Su(H)^5KR^ protein had a turnover similar to the wildtype protein (Appendix A). Apparently, these lysine residues are not essential or sufficient to mediate proteolysis of Su(H) in vivo in the presence of its binding partners. Perhaps, other lysine residues may be modified for targeting Su(H) for proteasomal degradation, in line with the idea that site-specificity of ubiquitination is rather promiscuous [59]. Accordingly, RBPJ protein is ubiquitinated at several sites in different contexts, by which proteasomal degradation is enhanced [59,62].

### 2.3. Su(H) Protein Levels Depend on the Presence of H

If H indeed has a limiting effect on the accumulation of Su(H) protein in the cells, then additional H protein might suffice to stabilise overexpressed Su(H) protein. In order to test this hypothesis directly, H was locally overexpressed along the antero–posterior boundary in wing imaginal discs, and the levels of Su(H) protein were monitored four hours after induction, i.e., at a time point when degradation is well advanced (see Figure 1). Fly stocks were generated, harbouring in addition the heat inducible constructs hs-Su(H)^wt^^-myc^ and hs-Su(H)^LLL^^-myc^, respectively. If H was able to protect ectopic Su(H) protein from degradation, we expected an enrichment of heat induced Su(H)^wt^^-myc^ protein in the same pattern of ectopically expressed H protein, which was indeed the case (Figure 3a). Heat-induced Su(H)^LLL^^-myc^, however, was barely detectable in the H-overexpression domain, indicating that this protein was degraded despite the presence of additional H protein (Figure 3b). These data underscore the idea that Su(H) is protected when in complex with its co-repressor H. Moreover, protection is not restricted to endogenous protein levels but rather correlates with the levels of H molecules, i.e., the more H protein that is available, the more Su(H) protein is stabilised. As their relationship is equimolar in the complex, the stabilisation of Su(H) by H is predicted to be linear, i.e., their levels rise or sink in parallel [23,24,44].

### 2.4. Su(H)–H Protein Interactions Are Short-Lived

Su(H) and H proteins form a 1:1 high affinity complex characterised by a remarkably strong binding energy within nanomolar range (~1 nM Kd) [23,24,37]. Yet, activated intracellular Notch NICD is able to displace H from repressor complexes, indicative of a rapid exchange of complex partners [23]. This suggests that the Su(H)–H protein interaction is transient, and that the partners might interchange between complexes or with newly translated molecules. In this case, we expect to affect existing Su(H)–H complexes when providing excess Su(H) protein, i.e., displacing endogenous by ectopic Su(H) molecules. If our hypothesis is correct that Su(H) protein is instable if solitary, the displaced endogenous Su(H) protein should be prone to degradation. In other words, exogenous and endogenous Su(H) are expected to compete for endogenous H. In contrast, Su(H)^LLL^, lacking the binding to H, should not affect Su(H)–H protein complexes at all. In order to test this hypothesis, we generated UAS-lines of UAS-Su(H)^wt^^-myc^ and UAS-Su(H)^LLL^^-myc^ at 96E by site-specific recombination, respectively, for a tissue specific overexpression with ptc-Gal4 as outlined above [53,54]. These lines were combined by genetic means with Su(H)^gwt-mCh^ [44]. In Su(H)^gwt-mCh^, the wildtype gene has been replaced by genome engineering with a copy encoding mCherry-tagged wildtype Su(H) protein [44]. Apart from the mCherry tag, Su(H)^gwt-mCh^ is indistinguishable from the wildtype [44]. Hence, in the recombined strains, the endogenous Su(H) protein can be detected by mCherry’s red fluorescence, while the overexpressed Su(H) protein can be monitored by the myc tag. In addition, UAS-Su(H)^LLL^^-myc^ was combined with Su(H)^LLL-mCh^ encoding the H-binding deficient Su(H) variant likewise labelled with mCherry [44], to allow assaying the influence of exogenous on the endogenous Su(H)^LLL^ protein levels. Figure 4 shows the results of the competition experiment. Remarkably, overexpression of Su(H)^wt^^-myc^ indeed resulted in the loss of endogenous Su(H)^gwt-mCh^ protein specifically within the overexpression domain, as predicted by our hypothesis (Figure 4a). We conclude that the overexpressed Su(H)^wt-myc^ protein displaces the endogenous Su(H)^gwt-mCh^ protein, which is then degraded. In contrast, overexpression of Su(H)^LLL-myc^ even caused an increase in endogenous Su(H)^gwt-mCh^ protein abundance (Figure 4b). This rather unexpected result might be explained by a proteasomal overload, caused by the preferential degradation of the overexpressed Su(H)^LLL-myc^ protein, leading to less degradation of the endogenous Su(H)^gwt-mCh^ protein. Overexpression of the Su(H)^LLL-myc^ variant in the Su(H)^LLL-mCh^ background showed a slight increase in the endogenous Su(H)^LLL-mCh^ protein levels as well (Figure 4c), which is in line with the overload hypothesis. If the high amounts of overexpressed Su(H)^LLL-myc^ protein saturated the proteasome, endogenous protein might be less prone to degradation, no matter whether wildtype or mutant. Note that the Su(H)^LLL-mCh^ protein localises preferentially within the cytoplasm as nuclear import requires the binding to H (Figure 4c) [29,44]. In addition, nuclear accumulation along the dorso–ventral boundary is apparent, i.e., at places of highest Notch activity, sparing Su(H)^LLL-mCh^ from degradation (Figure 4c) [29,44]. Taken together, these results demonstrate the protective effect of H protein on Su(H) degradation and the transient nature of their interaction at the same time.

### 2.5. Protection of Su(H) by Co-Activator and Co-Repressor Alike

Our experiments so far concentrated on the in vivo analysis of Su(H) stability within the repressor complex, i.e., when bound to H, since with Su(H)^LLL^ we have a mutation in hand that specifically abolished the Su(H)–H binding [24,44,63]. This mutant allowed us to measure Su(H)’s half-life in the absence of H. The analysis of Su(H) stability within the activator complex in vivo, however, is hampered by the lack of a likewise specific mutation affecting Su(H)–Notch binding only. Hence, we turned to HeLa cells as the human genome lacks a H homologue, known to be restricted to the arthropods [18,61]. Moreover, biologically active Notch, i.e., NICD is expected at low levels. To avoid any influence of the homologous RBPJ protein, HeLa RBPJ^KO^ cells were used for the analysis. In these cells, a fragment of the RBPJ gene was deleted by the CRISPR-Cas9 technology, resulting in a complete loss of RBPJ protein expression [29].

The RBPJ-deficient HeLa RBPJ^KO^ cells were first transfected with a Su(H)^wt-myc^ expression construct. Thereafter the cells were treated with cycloheximide (CHX) to stop translation, and protein extracts were prepared at different time points after treatment (0 h, 1 h, 2 h, 4 h and 6 h) to be evaluated and quantified in Western blots (Figure 5a,c and Appendix A). Confirming the data obtained for Su(H)^LLL-myc^ in the in vivo experiments in fly tissue, a half-life of about two hours was determined for Su(H)^wt-myc^ protein in HeLa RBPJ^KO^ cells (Figure 5a,c), matching the half-life of RBPJ protein determined in HEK293 cells [39]. If our hypothesis is correct that Su(H) is stable when within complexes, we predicted an accumulation of the transfected Su(H)^wt-myc^ protein in the presence of ectopic H or NICD protein in HeLa cells as well. To avoid a cellular response by the expression of NICD, we used ICN^RAMANK^ bearing the entire CSL binding domain, however, lacking the transactivation domain of NICD [2,14,23]. Care was taken in these experiments to ensure that the expression constructs are present in an equimolar ratio. Indeed, upon co-expression of either wildtype H or ICN^RAMANK^, Su(H)^wt-myc^ remained stable at levels of about 70% of the induced protein (Figure 5a,c, Appendix A). Interestingly, stabilisation of Su(H)^wt-myc^ appeared independent of the cofactor, i.e., Su(H)^wt-myc^ protein levels were similar, whether H or ICN^RAMANK^ was used, suggesting that either protects Su(H) from degradation equally well. This result supports the notion that the complex formation protects solitary Su(H) from degradation, since both activator and repressor complexes assemble with similar binding affinity [23,25,37].

In a further experiment, Su(H)^LLL-myc^ was transfected instead of the wildtype protein. Indeed, the stability of Su(H)^LLL-myc^ protein matched that of Su(H)^wt-myc^ in the absence of cofactors (compare Figure 5a–d). This result was expected since Su(H)^LLL-myc^ should exist solitary in RBPJ^KO^ HeLa cells just like Su(H)^wt-myc^ lacking respective binding partners. Accordingly, Su(H)^LLL-myc^ protein was stabilised by ICN^RAMANK^, but not by H, which is expected based on the lack of H but not ICN-binding (Figure 5b,d). These results clearly demonstrate that Su(H) protein, bound to either H or ICN^RAMANK^, is stabilised in repressor and activator complexes also in HeLa cells (Figure 5).

### 2.6. RBPJ Might Be Stabilised by Cofactors as well

Reports on the half-life of RBPJ protein vary dramatically depending on cell type and setting. For example, a rapid turnover of RBPJ within 2 h, dependent on Psn2 and p38 MAPK, was reported in HEK293 cells [39], matching the stability of solitary Su(H) protein in *Drosophila* tissue and HeLa cells, respectively (see Figure 1 and Figure 5). In contrast, a half-life of ten hours or more was measured for RBPJ in other cell types (human umbilical vein endothelial cells, U251 glioma cells) even upon induction of proteolysis by VEGF treatment or by metabolic stress [38,40]. We used the RBPJ-mutant HeLa RBPJ^KO^ cells to address the turnover of wildtype human and murine RBPJ proteins that both were very stable with a half-life beyond 8 h (Figure 6a,b,e,f and Appendix A). Apparently, RBPJ protein is protected from degradation most likely by one of its many binding partners [16,37]. We have shown earlier that murine Rbpj behaves very similarly to the fly orthologue, both with regard to stability and nuclear entry in fly tissue [29,30,44,64]. For example, the H-binding deficient Rbpj^LLL^ variant had a similar turnover to its fly homologue Su(H)^LLL^ [44,64]. Structural analyses, however, have revealed that H-binding to CSL proteins differs from that of mammalian co-repressors (reviewed in: [16,37]). Whereas H binds specifically to the C-terminal domain of Su(H), thereby distorting the overall structure and precluding Notch-binding, other co-repressors bind preferentially or exclusively to the beta-trefoil domain, competing with the primary contacts of the Notch RAM-domain [16,22,23,24,37,60,65,66,67].

In order to address the impact of co-repressors on Rbpj’s stability in HeLa cells, we generated Rbpj mutants affecting co-repressor binding. Structural information on Rbpj repressor complexes is available for several co-repressors, including SHARP, RITA1 and KyoT2/FHL2 [22,37,66,67,68], that are all expected to be expressed in HeLa cells. Mutants were designed within the beta-trefoil and C-terminal domains of Rbpj according to the literature in order to impinge the interactions with either RITA, SHARP or Notch [37,69,70]. Two constructs were generated with three and five amino acids exchanged, respectively, namely Rbpj^FAL^ (F261A, A284V, L388A) and Rbpj^EEFAL^ (E259A, E260A, F261A, A284V, L388A). They were confirmed by yeast two-hybrid assays to prevent binding to SHARP^RBPID^ and to RITA1, and to affect the binding to Notch RAMANK as well (Appendix A). The stability of the two mutant variants, Rbpj^FAL^ and Rbpj^EEFAL^, was then assayed in HeLa RBPJ^KO^ cells. Unexpectedly, Rbpj^FAL^ was indistinguishable from the wildtype and remained stable upon translation arrest (Figure 6c,f and Appendix A), despite its lack of binding to RITA and SHARP^RBPID^ (Appendix A). Rbpj^EEFAL^, however, was slightly degraded over time (Figure 6d,f and Appendix A), suggesting that the impaired binding to Notch RAMANK affected its stability. Overall, Rbpj protein appears largely stable in HeLa cells, presumably because the mutants are still able form complexes with one or several of the many known RBPJ partners [15,16,37].

### 2.7. Stability of Su(H) Is Regulated by MAPK-Dependent Phosphorylation

Previous work indicated that degradation of the short-lived RBPJ protein in HEK293 cells is promoted by a p38 MAPK-dependent phosphorylation [39]. The phosphorylation occurs at threonine 339 and is mediated by Presenilin 2 and the p38 MAPK δ (MAPK13) [39]. Regulated protein degradation entails a specific recognition of the substrate destined for proteolysis. In fact, many substrates acquire their degron by secondary modification, typically by phosphorylation [71], allowing for specific, temporal and/or spatial control of protein abundance. Presumably, phospho-T339 serves as a typical degron in RBPJ, raising the possibility of a likewise degron in Su(H) at the conserved position threonine 426.

We have shown before in overexpression experiments that phospho-site mutations at T426 affected Su(H) activity [72,73]. Two mutants were investigated, a knock-out of the phospho-site, Su(H)^MAPK-ko^ with a T426A replacement, and a phospho-mimetic variant T426E Su(H)^MAPK-ac^. Whereas the phospho-deficient Su(H)^MAPK-ko^ mutant provoked a stronger Notch signalling activity, the Su(H)^MAPK-ac^ was clearly impeded in the Notch response [72,73]. In order to address the possibility that the modulation of Notch activity is a result of Su(H) protein availability, we cloned both phospho-site mutants under the control of the heat shock promoter and generated transgenic fly lines allowing to monitor protein abundance over time upon transient induction. Corresponding to the experiments described above (Figure 1), four hours after protein induction, the wildtype Su(H) signals had dropped to about a 67% level of the maximum protein induction (compare Figure 1 and Figure 7a,b). A similar value was observed for the phospho-mimetic Su(H)^MAPK-ac^ variant, whereas the phospho-deficient Su(H)^MAPK-ko^ mutant appeared unchanged (Figure 7a,b). We interpret this result as an indication that a phosphorylation on T426 is a pre-requisite for the degradation of the Su(H) protein in vivo. The phospho-mimetic isoform Su(H)^MAPK-ac^, however, was not more labile than the wildtype protein, suggesting that it is protected from degradation by its binding partners. In this case, the phosphorylation site may only be accessible in the solitary Su(H) protein, and otherwise protected by the protein complexes interfering with MAPK access.

## 3. Discussion

Despite the fact that CSL represents the core transcription factor of canonical Notch signalling, mechanistic insights on its own regulation are scarce. In contrast to other Notch signalling components that are massively modified post-translationally, CSL is primarily phosphorylated [8,74]. Phosphorylation is a powerful tool to rapidly change the activity of a protein. *Drosophila* Su(H) is phosphorylated in the N-terminal domain at serine 269, impeding its DNA binding affinity in the context of hematopoiesis [75,76]. In this case, phosphorylation directly affects Su(H) function as a transcriptional regulator, even when sitting on the DNA within activator or repressor complexes [75,76]. Hence, Su(H) phosphorylation at serine 269 directly interferes with ongoing Notch signalling and is predicted to be a target for a stress response requiring immediate break of Notch activity. In addition, Su(H) is phosphorylated at threonine 426 by MAPK, just like its human counterpart RBPJ in HEK293 cells [39,72,73]. This post-translation modification, however, generates a phospho-degron in RBPJ causing rapid proteasomal degradation. Our results demonstrate that a phospho-deficient mutant Su(H)^MAPK-ko^ protein is largely stable in vivo, indicating that this mechanism of MAPK-dependent destruction of CSL is conserved between vertebrates and invertebrates. In contrast, a phospho-mimetic Su(H)^MAPK-ac^ mutant displayed attenuated Notch activity, conforming to a lowered abundance of Su(H) in response to MAPK activity [72,73]. Activation of the EGFR receptor eventually leads to an activation of MAPK, which then may interfere with CSL stability. In fact, an antagonistic relationship between Notch and EGFR signalling has been observed in flies as well as in mammals, which might in part result from CSL modification and degradation (reviewed in: [77,78,79]). RBPJ degradation as a mechanism of Notch pathway regulation has been uncovered several times in cancer cells. For example, metabolic stress in glioma cells resulted in the induction of cyclin F in a FOXO1-dependent manner that leads to RBPJ proteolysis. As one of RBPJ’s target genes is often mutated in glioma, its downregulation diminished the malignant phenotype [40]. Similarly, in bladder cancer cells, overexpressed RITA1 induced degradation of RBPJ by recruiting the E3 ligase TRIM25 [80]. In this case, however, RBPJ proteolysis appeared to increase malignancy, which may apply for other cancer types linked to TRIM25 activity as well [80,81,82,83].

Interestingly, under normal circumstances, CSL is largely stable in many cell types and contexts, however, it is subjected to degradation in certain cancer cells. The fact that dysregulation of RBPJ’s stability is strongly linked to cancer development is an argument for the need of a tight control of RBPJ abundance. Steady-state levels of the Notch receptor as well as its ligands are maintained by several mechanisms, thereby regulating the stoichiometry of receptor–ligand interactions (reviewed in [1,2,84]). Moreover, the transcriptional activation process mediated by CSL is accompanied by the proteasomal destruction of NICD [51,85]. During this process, the co-activator Mam promotes hyper-phosphorylation of a conserved degron located within the C-terminal domain of NICD, which is then recognised by the F-box ubiquitin ligase Fbw-7/SEL-10, resulting in proteasomal degradation [85]. In fact, transcription-coupled proteolysis is a hallmark to unstable transcriptional activators, coined the ‘black widow’ model [86]. In addition to coupling transcriptional activation to destruction, the proteasome has a further role in transcriptional activation and elongation by remodeling chromatin landscapes. This involves several subunits of the regulatory 19S cap of the proteasome, as well as the E3 ubiquitin ligase Bre1 [86,87]. Interestingly, the co-repressor H interacts directly with a subunit of the regulatory 19S cap, linking Su(H) to chromatin regulation as well [88]. In fact, Notch target gene expression is particularly susceptible to ubiquitin-dependent chromatin activation by BreI [89]. Notably the co-repressor complexes that assemble around CSL contain a variety of chromatin-modifiers regulating promoter and enhancer accessibility of Notch target genes (reviewed in [15,19]).

Our work demonstrates that *Drosophila* Su(H) is protected from degradation by binding to its partners H or NICD. In fact, our experiments failed to uncover any quantitative differences between the two in protecting Su(H). Biochemical analyses have shown that both proteins bind Su(H) with a similar affinity within nanomolar range, suggesting similar dynamics in complex formation and dissociation [20,23,25]. We cannot exclude, however, the possibility that it does not matter who binds to Su(H), if protection results from simply masking the presumptive degron against recognition by MAPK and E3-ligases, respectively. The MAPK target site is located at an exposed position at the very start of CSL’s C-terminal domain. Binding of NICD, however, presumably precludes accessibility [14,60,65]. Likewise, the binding of H is expected to do so, firstly by distorting the structure of the C-terminal domain, and secondly by its large protein size [24,90]. Apart from SHARP that also binds to the C-terminal domain of CSL, the other mammalian co-repressors are thought to exclusively contact the beta-trefoil domain in a manner similar to NICD [16,19,22]. Depending on their bulkiness and additional cofactors, the latter co-repressors may not hinder the access of MAPK, and hence the formation of a phospho-degron accompanied by CSL’s destruction. In fact, co-repressor binding may contribute little to Rbpj turnover, since the mutant Rbpj^FAL^ protein is as stable as the wildtype despite its lack of binding to either SHARP^RBPID^ or RITA1 (Figure 7 and Appendix A). The second mutant tested, however, Rbpj^EEFAL^ appeared less stable, having a conspicuously reduced affinity to NICD as well (Figure 7 and Appendix A). These data suggest that the binding to NICD, rather than to co-repressors, might primarily protect Rbpj from degradation. We favour the idea, however, that further cofactors present in HeLa cells prevented Rbpj proteolysis. The many possible interaction partners would easily explain the remarkable stability of mammalian CSL protein, as well as the drastic differences in half-life measurements [38,39,40,80]. Interestingly, the opposite mechanism was discovered for human Notch1 [91]. The intracellular domain of the Notch1 receptor NICD1, but none of its paralogues contain an N-terminal degron, which results in rapid degradation if not bound to CSL. In this case, CSL protected NICD1 from proteolysis in order to dampen stochastic flux in Notch1 signal transduction [91].

Overall, out data provide evidence for a conserved mechanism for the regulation of CSL turnover, involving the generation of a MAPK-dependent phospho-degron on the one hand, and the protection from degradation by CSL-cofactor binding on the other hand. It can be assumed that CSL protein turnover is critically relevant to Notch signalling in many contexts. If complex formation protected CSL from degradation, unbound CSL should disappear rapidly, precluding spurious activation of Notch target genes. The dynamics of CSL-cofactor interactions eventually contribute to the fine-tuning of Notch signalling output under normal circumstances and may result, e.g., in cancer development if disturbed.

## 4. Materials and Methods

### 4.1. Generation of Su(H) and RBPJ Constructs

A myc tag was added to Su(H)^wt^ and Su(H)^LLL^ cDNA in pBT (Stratagene) by first mutating the stop codon into a *Bam* HI site by PCR amplification, ligating into pBT, followed by opening with *Bam* HI/*Xba* I to introduce annealed primers with respective overhangs encoding the myc tag. The constructs were subsequently shuttled via *Eco* RI/*NotI* I into pUAST-attB [49] and phs-attB (see below). Amino acid replacement mutants were generated in Su(H)–myc in pBT or pMT (pRmHa-3) [92] by site directed mutagenesis using the Q5^®^ site-directed mutagenesis kit (New England Biolabs). In detail, Su(H)^K389/K393A-myc^ was generated using Su(H)^wt-myc^ in pBT in two steps (first K389A, then K393A), to be eventually shuttled into phs-attB. Likewise, Su(H)^K5A-myc^ was mutated stepwise by changing K103/104A and eventually K112A on top of K389/393A. Su(H)^K5R-myc^ was also generated stepwise by first replacing K343/359R, second K200R and finally K389/393R on top. To generate phospho-site mutations, Su(H)^wt-myc^ was transferred via an *Eco* RI/*Acc65* I digest into pMT vector, and subsequently the internal 0.8kb *Bsm* I/*Bst1107* I fragment replaced by the respective fragments from pUASt-Su(H)^MAPK-ko^ and pUASt-Su(H)^MAPK-ac^, respectively [72], to be shuttled back into phs-attB.

The vector phs-attB was generated by the excision of the 0.58kb heat shock promoter (HSE) from pCaSpeR-hs-RX8 [48,90] with *Eco* RV/*Eco* RI digest, cloning into likewise-opened pBT to yield pBT-HSE, and by replacement of the UAS-elements from pUAST-attB [49] with the respective fragment from pBT-HSE after *Hind* III/*Eco* RI digest. Su(H) wildtype and respective mutant variants, fused C-terminally to myc, were integrated as *Eco* RI/*Not* I fragments into phs-attB.

pcDNA3.1-Su(H)^wt-myc^ and the mutants Su(H)^LLL-myc^, Su(H)^MAPK-ko-myc^, and Su(H^)MAPK ac-myc^, respectively, were transferred from respective phs-attB subclones as *Eco* RI/*Not* I fragments into pcDNA3.1 (Invitrogen). To generate pcDNA3.1-H, the cDNA encoding full length H was shuttled as 3.3kb *Kpn* I fragment from pMT-H [29,93]. pcDNA-ICN-RAMANK was constructed by shuttling pESC-RAMANK [23] as *Spe* I/*Sac* I fragment into pBT, from there as *Eco* RI/*Sac* I fragment into pMT to be inserted into pcDNA3.1 as *Eco* RI/*Xba* I fragment.

The clone pcDNA3.1 hRBPJ2N expressing human RBPJ was a gift from F. Oswald (University of Ulm, Germany) [94]. Murine Rbpj in pGEX6P1 was kindly obtained from Rhett A. Kovall (University of Cincinnati, USA) [25]. It was PCR amplified to allow subcloning as *Eco* RI/*Bam* HI fragment into pBT-myc to yield C-terminally tagged Rbpj^myc^. Stepwise mutagenesis of Rbpj^myc^ was performed by Q5^®^ site-directed mutagenesis kit (New England Biolabs), first F261A, second A284A, and third E259A/E260A. L338A was mutated separately in Rbpj^myc^, to eventually exchange the wildtype C-terminal 0.68kb *Afl* II/*Not* I fragment in the double mutant F261A-A284V and quadruple mutant E259A-E260A-F261A-A284V, respectively with the L388A mutant. The mutant Rbpj constructs were subsequently shuttled as *Eco* RI/*Not* I fragments into pcDNA3.1. All constructs were confirmed by diagnostic digests and PCR; they were all sequence verified. Mutagenesis primers are listed in the Appendix A.

### 4.2. Fly Work

#### 4.2.1. Husbandry and Transgenic Lines

Transgenic lines were established by PhiC31 mediated site-specific recombination at the identical position in 96E using the ΦX-96E strain as described before [23,49]. Thereby, any position effects by varying chromosomal environments were circumvented. Fly stocks were maintained at 18 °C on standard corn meal fly food; crosses were kept at 25 °C to speed up development. Stocks were kept at low density to avoid overcrowding in all experiments. Combinations of hs-Su(H)* variants with UAS-H at 68E [23] or UAS-DTS5 [51] and ptc-Gal4 [54], or UAS-Su(H)^wt-myc^ and UAS-Su(H)^LLL-myc^ variants with the genome engineered Su(H)^gwt-mCh^ and Su(H)^LLL-mCh^, respectively, were performed by standard genetics methods. Dominant markers on balancer chromosomes were used to select the desired genotypes; mutant larvae were recognised by the lack of a ubiquitously expressed GFP marker contained on the balancer chromosomes as outlined before [44]. Genotypes of transgenic lines and established stocks were verified by PCR, followed by diagnostic digests where appropriate.

#### 4.2.2. Heat Shock Regimen

Third instar larvae were subjected to a 30 min heat shock by shifting vials from 18 °C into a 37 °C warm water bath. Care was taken that the vials were completely submerged in water. Heat was stopped by shifting the vials back to 18 °C. Larvae were dissected at the times after heat shock indicated, and imaginal discs stained according to standard protocols as outlined before [64,95]. To ensure identical conditions during antibody staining, master mixes of primary and secondary antibodies were used for all the discs from the same experiment.

#### 4.2.3. DTS5 Experiment

For the DTS5 experiment, stable lines combining hs-Su(H)* variants with UAS-DTS5 [51] were established and verified. About six males were crossed with about ten ptc-Gal4 virgin females to avoid overcrowding. Crosses were transferred to fresh food daily to synchronise egg deposition, and then reared at 25 °C. Three days thereafter, vials were shifted to 29 °C for 24 h, and then subjected to a 30 min heat shock at 37 °C, to be shifted back to 29 °C for another 8 h. Control larvae were kept for the same time at 29 °C without heat shock. Larvae were collected at 1 h, 2 h, 4 h and 8 h after heat shock and subjected to antibody staining as above.

#### 4.2.4. Immunocytochemistry of Fly Tissue

The following antibodies were used for staining of the imaginal discs: mouse anti-myc 1:500 (Cell Signaling Technology, Frankfurt, Germany), guinea pig anti-H 1:250 [96], rabbit anti-mCherry 1:500 (GTX128508, GeneTex, obtained from Biozol, Eching, Germany); secondaries coupled to FITC or Cy3 were used at 1:200 (Jackson Immuno-Research, obtained from Dianova, Hamburg, Germany). Imaginal discs within the same plane were chosen, and five 1 µm thick sections taken at identical confocal settings using a Zeiss Axioskop coupled with a BioRad MRC1024 confocal microscope, using LaserSharp 2000 software (Carl Zeiss, Jena, Germany). For the quantification of signal intensity, sections were stacked, and the mean gray value of the entire disc recorded using ImageJ. Measurements of ten discs were sampled, the mean and SD (standard deviation) are indicated. Statistical analyses were performed by ANOVA for multiple comparisons, using a two-tailed Dunnett’s approach.

### 4.3. Cell Culture Experiments

CRISPR edited HeLa RBPJ^KO^ cells do not express RBPJ (clones #42 and #4.42) [29]; they were obtained from F. Oswald and T. Borggrefe (Universities of Ulm and of Giessen, respectively). They were grown in Dulbecco’s modified Eagle’s medium (DMEM, Gibco/Thermo Fisher Scientific, Waltham, MA, USA) supplemented with 10% FBS and 1% penicillin/streptomycin. For transfection we used Lipofectamine 2000 transfection reagent (Invitrogen/Thermo Fisher Scientific, Waltham, MA, USA) according to the manufacturer’s instructions. For the analysis of protein stability over time, 3 × 10^5^ HeLa RBPJ^KO^ cells were seeded on 60 mm cell culture plates. After 24 h a total of 6 µg of DNA from the respective pcDNA3-constructs was transfected in an equimolar ratio. The cells were harvested 18 h later by a Trypsin-EDTA treatment, washed in PBS (pH 7.4), and dissolved in 5.5 mL DMEM with 10% FBS (BIO & SELL, Nürnberg, Germany). Then, each 1 mL of the cell suspension was seeded in one well of a 12-well cell culture plate to allow analysis of the five different time points, ensuring the same transfection efficiency for each. After another 24 h cells were treated with 50 µg/mL Cycloheximide (CHX; Sigma-Aldrich (C7698-1G). For Western blot analyses, the cells were lysed before CHX treatment and at 1 h, 2 h, 4 h and 6 h after CHX treatment in 75 µL binding buffer (20 mM HEPES pH 7.6, 150 mM KCl, 2.5 mM MgCl_2_, 10% glycerol, 0.05% NP-40, 1 mM DTT, 1 complete ULTRA protease inhibitor mini tablet (Roche, Basel, Switzerland), and frozen in 20 µL aliquots at −80 °C. Aliquots were mixed with 10 µL loading dye and boiled before loading half of the total on 10% SDS gels for protein separation during PAGE. The Western blots were sliced accordingly to detect respective proteins in parallel, and probed with mouse anti-myc Tag 9B11 1:1000, rabbit anti-RBPSUH 1:500 (both from Cell Signaling Technology, Frankfurt, Germany), guinea pig anti-H 1:500 [96], and mouse anti beta-Tubulin A7 1:1000 (developed by M. Klymkowsky; obtained from *DSHB*, IA, USA). Goat secondaries, coupled to alkaline phosphatase, were used 1:1000 for detection (Jackson Immuno-Research, obtained from Dianova, Hamburg, Germany). Quantification was performed with the ImageJ gel analysis program on n = 5 blots if not stated otherwise. Beta-tubulin signals were used as internal standard. Statistical analyses were performed by ANOVA for multiple comparisons, using a two-tailed Dunnett’s approach.

### 4.4. Yeast Two-Hybrid Experiments

Yeast two-hybrid assays, based on the Golemis–Brent hybrid system, were performed for protein–protein interaction analyses as outlined earlier, using EGY48 yeast cells [64,97]. Rbpj constructs were cloned in pJG4-5 providing the transactivation domain. To this end, wildtype Rbpj, Rbpj^FVL^ and Rbpj^EEFAL^ were shuttled from pcDNA3.1 by *Eco* RI/*Xho* I digests. Interactions were investigated with pEG202 vector constructs providing the LexA DNA binding: pEG-H-NTCT [23], pEG-RITA1 [98], pEG-SHARP^RBPID^-GFP [22], pEG-NICD [93,99]. Expression was verified by Western blots with protein extracts from transformed yeast cells, using rabbit anti-LexA antibodies (1:3000; BioAcademia, Osaka, Japan) for the pEG constructs, and rat anti-HA antisera (1:500, Roche, Basel, Switzerland), for the pJG-constructs, respectively. AP-coupled goat secondaries were used for Western blots at 1:1000 dilution as outlined above. Experiments were performed twice and in triplicate.

## Figures and Tables

**Figure 1 ijms-23-12336-f001:**
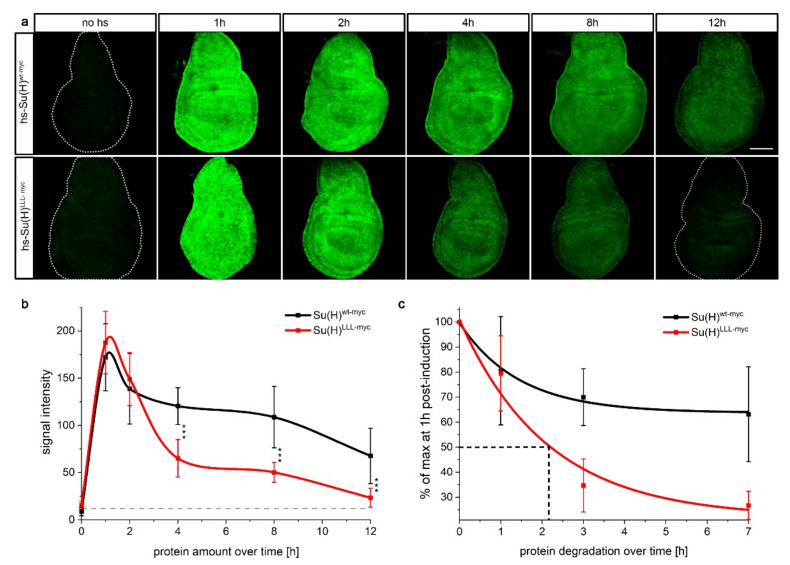
In vivo stability of Su(H) protein in imaginal tissue. (**a**) Su(H)^wt-myc^ and Su(H)^LLL-myc^ protein expression, respectively, induced by a 30 min heat pulse, and monitored via the myc tag in wing imaginal discs at specific time points after heat shock as indicated. Typical examples are shown. Size bar, 100 µm in all panels. (**b**) Quantification of Su(H)^wt-myc^ and Su(H)^LLL-myc^ protein amounts. Signal intensity was recorded, and data sampled from ten independent experiments each performed in parallel under identical conditions. SD is indicated. The grey dashed line denotes the baseline, i.e., signal intensity without heat shock. Note similarity in induction, followed by a rapid drop at 2 h for both protein variants. Whereas Su(H)^LLL-myc^ protein amounts sink rapidly and significantly, Su(H)^wt-myc^ levels are stabilised at about 67% of the maximum induction. *** *p* < 0.001. (**c**) Drop of protein levels over time; peak expression at 1 h after heat shock induction was taken as 100%. Note stabilisation of Su(H)^wt-myc^ at about 67% max. Half-life of Su(H)^LLL-myc^ is at around 2 h.

**Figure 2 ijms-23-12336-f002:**
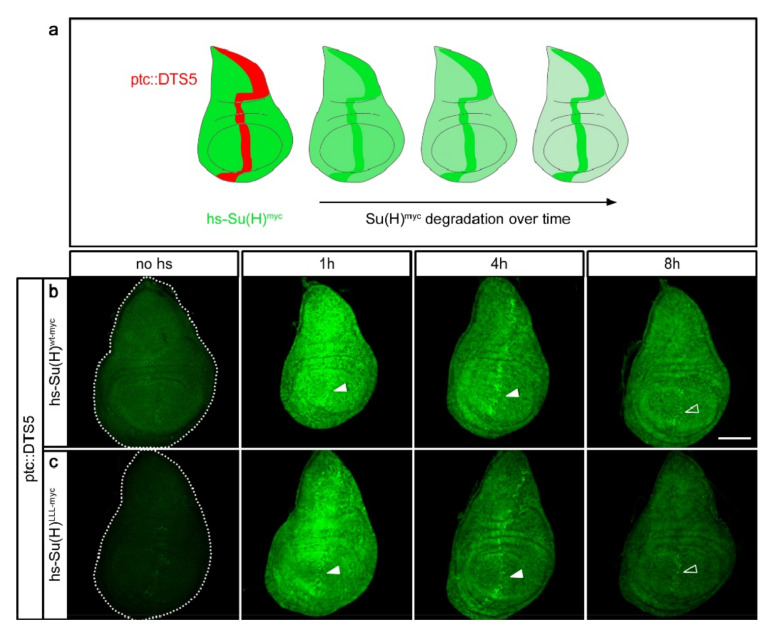
Su(H) is subjected to proteasomal degradation. (**a**) Scheme of the experimental design [51] showing cartoons of wing imaginal discs. Proteasomal activity is inhibited specifically along the antero–posterior boundary by overexpressing the DTS5 mutant in the ptc-pattern at restrictive temperature, indicated by a red stripe (ptc::DTS5). The protein of interest, shown in green, is induced ubiquitously by a short heat pulse. It is degraded over time except in the domain of proteasomal malfunction (green stripe). (**b**) hs-Su(H)^wt-myc^ and (**c**) hs-Su(H)^LLL-myc^ protein expression in the ptc::DTS5 background, monitored over time as indicated. Note the enrichment of Su(H) protein along the antero–posterior boundary, indicating stabilisation in the absence of proteasomal activity (arrowheads). Size bar, 100 µm in all panels. Genotypes are: ptc-Gal4/UAS-DTS5; hs-Su(H)^wt-myc^/+ (**b**) and ptc-Gal4/UAS-DTS5; hs-Su(H)^LLL-myc^/+ (**c**).

**Figure 3 ijms-23-12336-f003:**
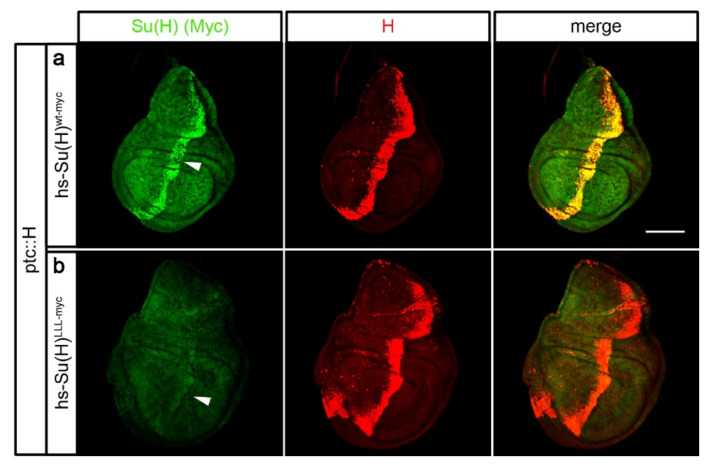
Su(H) protein is protected from degradation by H protein in vivo. (**a**) The local overexpression of H protein (ptc::H) is sufficient to stabilise Su(H)^wt-myc^ protein (arrowhead) induced by a heat pulse. (**b**) In contrast, Su(H)^LLL-myc^ protein disappears over time (arrowhead). Imaginal discs were stained 4 h after Su(H) protein induction specifically with anti-myc antibodies (green), and for H protein (red), respectively. Size bar, 100 µm. Genotypes are: ptc-Gal4/+; hs-Su(H)^wt-myc^/UAS-H (**a**) and ptc-Gal4/+; hs-Su(H)^LLL-myc^/UAS-H (**b**).

**Figure 4 ijms-23-12336-f004:**
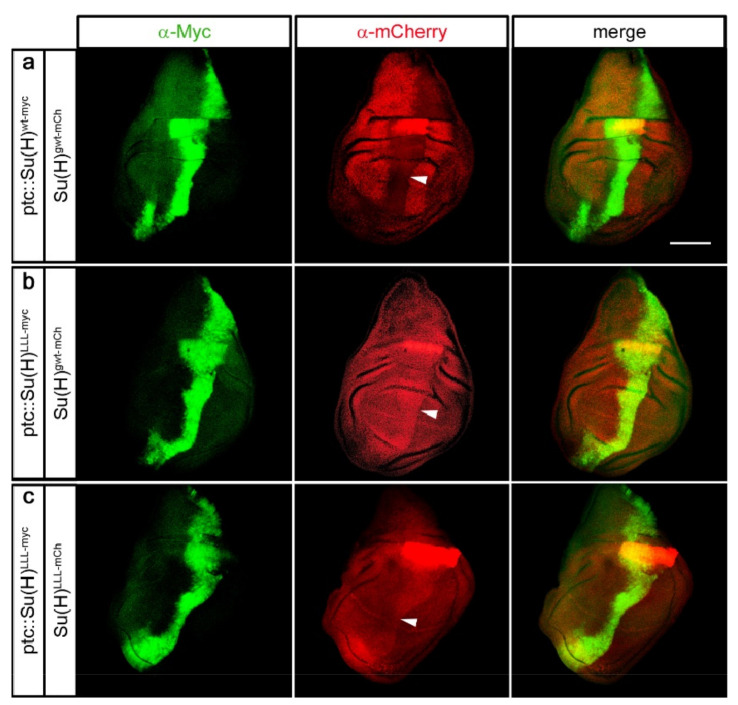
Competition between endogenously and ectopically expressed Su(H). (**a**–**c**) Expressions of UAS-Su(H)^wt-myc^ and Su(H)^LLL-myc^ constructs were locally induced along the antero–posterior boundary with ptc-Gal4 as indicated, and detected specifically with anti-myc antibody staining (green). Endogenous Su(H) protein expression was visualised by mCherry antibody staining (red) [44]. (**a**) Overexpression of wild type Su(H)^wt-myc^ results in the disappearance of the endogenous Su(H)^gwt-mCh^ protein (arrowhead). (**b**) In contrast, overexpression of H-binding deficient Su(H)^LLL-myc^ causes enrichment of endogenous Su(H)^gwt-mCh^ protein (arrowhead). (**c**) A similar, albeit weaker enrichment is observed for Su(H)^LLL-mCh^ upon overexpression of Su(H)^LLL-myc^. Arrowheads mark the position of the antero–posterior boundary in (**a**–**c**). Size bar, 100 µm. Genotypes are: ptc-Gal4 Su(H)^gwt-mCh^/Su(H)^gwt-mCh^; hs-Su(H)^wt-myc^/+ (**a**), ptc-Gal4 Su(H)^gwt-mCh^/Su(H)^gwt-mCh^; hs-Su(H)^LLL-myc^/+ (**b**), ptc-Gal4 Su(H)^LLL-mCh^/Su(H)^LLL-mCh^; hs-Su(H)^LLL-myc^/+ (**c**).

**Figure 5 ijms-23-12336-f005:**
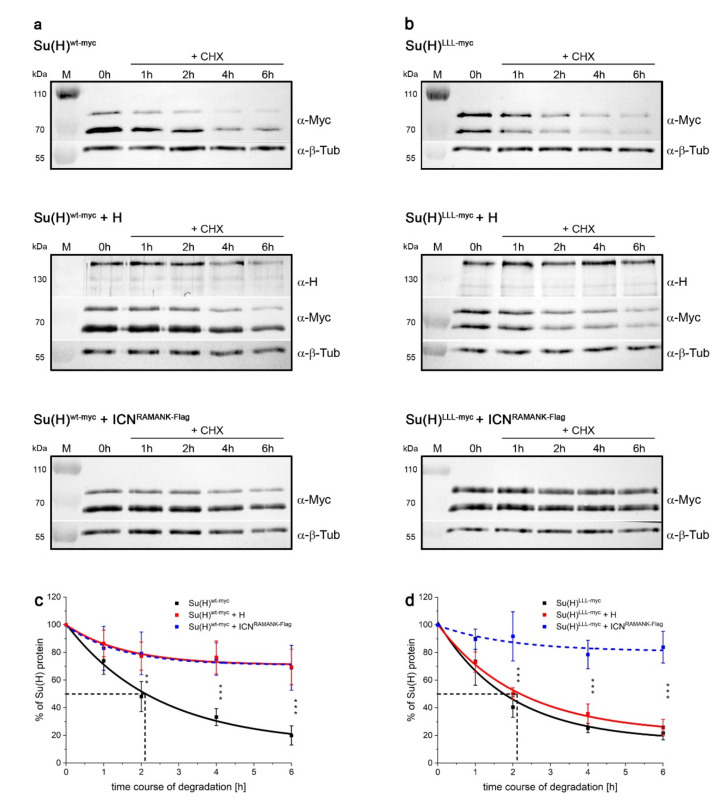
(**a**) Su(H)^wt-myc^ protein and (**b**) Su(H)^LLL-myc^ protein expressed in HeLa RBPJ^KO^ cells, in the presence or absence of either co-repressor H or co-activator ICN^RAMANK^ as indicated. Translation was inhibited by addition of cycloheximide (CHX), and proteins extracted thereafter at the indicated time points. Su(H) was detected with anti-myc antibodies. Western blots were cut to monitor respective protein expression simultaneously with adequate primary antibodies as indicated. Beta-Tubulin (ß-Tub) served as internal control for quantification. Representative Western blots are shown. (**c**,**d**) Quantification of Su(H) protein levels over time in the presence or absence of either H or ICN^RAMANK^, relative to beta tubulin, from five independent experiments performed under identical conditions. SD is shown; significant differences are observed relative to Su(H)^wt-myc^ (**c**) or to Su(H)^LLL-myc^ with or without H (**d**) (*** *p* < 0.001; ** *p* < 0.01).

**Figure 6 ijms-23-12336-f006:**
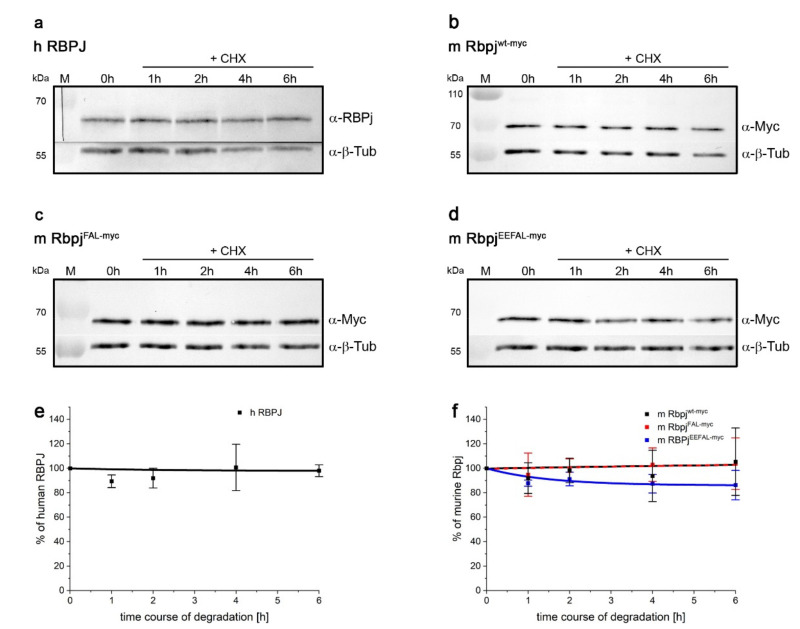
Stability of human RBPJ and mouse Rbpj proteins in HeLa RBPJ^KO^ cells. Human RBPJ^myc^ protein (**a**), mouse Rbpj^wt-myc^ protein (**b**), Rbpj^FAL-myc^ (**c**) and Rbpj^EEFAL-myc^ (**d**), expressed in RBPJ^KO^ HeLa cells. Translation was inhibited by addition of cycloheximide (CHX), and proteins extracted thereafter at the indicated time points. Human RBPJ was detected with anti-RBPSUH antibodies, and murine Rbpj variants based on the myc tag. Western blots were cut to monitor respective protein expression simultaneously with anti-beta-Tubulin (β-Tub) for an internal standard. Representative Western blots are shown. (**e**,**f**) Quantification of human RBPJ protein (**e**), and murine Rbpj protein variants (**f**) relative to beta-tubulin from three independent experiments performed under identical conditions.

**Figure 7 ijms-23-12336-f007:**
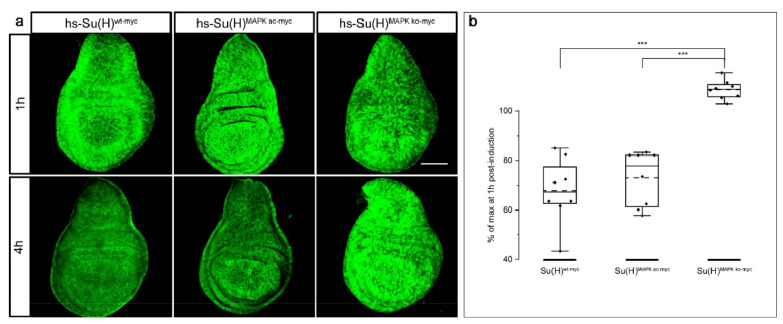
Mutation of the MAPK site precludes Su(H) protein degradation. (**a**) Expression of hs-Su(H)^wt-myc^, hs-Su(H)^MAPKac-myc^ and hs-Su(H)^MAPKko-myc^ protein, respectively, induced by a 30 min heat pulse, and monitored via the myc tag in wing imaginal discs at 1 h and 4 h after heat shock as indicated. Typical examples are shown. Size bar, 100 µm in all panels. (**b**) Quantification of eight independent experiments reveals statistically highly significant enrichment of phospho-deficient Su(H)^MAPK-ko^ protein (*** *p* < 0.001). Full centre lines show medians, dashed lines the average, box limits indicate the 25th and 75th percentiles, whiskers extend the interquartile range by 1.5-fold.

## Data Availability

Not applicable.

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
