# Peer review of "The Binding of CSL Proteins to Either Co-Activators or Co-Repressors Protects from Proteasomal Degradation Induced by MAPK-Dependent Phosphorylation"

_ijms, 2022, doi:10.3390/ijms232012336_

Round 1
Reviewer 1 Report
This manuscript is very well-written. The experimental design is in logic order. The authors have provided detailed discussions for their results, and the conclusions are reasonably built upon on experimental evidence. The reviewer has only a couple of minor suggestions:
1. section 2.2: it is not very clear how this experiment was conducted. For example, the authors state that DTS5 prevents proteasome function at temperature above 29C, then did this mean that the authors perform the experiments at elevated temperature or not? Also, in line 204, it is not clear if one of the protein is degraded (since either..or...) or both of them were deactivated (which is a neither... nor... relationship).
2. The authors rely heavily on fluorescence microscopy for determining the quantity and degradation of the studied proteins. While degradation of proteins could be the main reason of reduced fluorescence intensity, the bleaching of fluorescence probe could also contribute to such intensity decrease. It is recommended that for future studies, other biophysical characterization methods can also be used to provide more insights into the conformational change, degradation, aggregation behavior of proteins.
Author Response
We thank the Reviewer for the insightful comments. We accounted for the suggestions and
- We reworded section 2.2 to:
To allow for a tissue-specific overexpression of DTS5, the Gal4-UAS system was employed: UAS-DTS5 combined with ptc-Gal4 was expressed along the antero-posterior boundary of imaginal discs [53-54]. After rearing the larvae at permissive temperature, they were shifted for a day to 29oC to block proteasomal activity before applying the heat shock. In this background, accumulation of heat-induced Su(H)wt-myc or Su(H)LLL-myc protein was examined over time (Figure 2a). As outlined above, ectopic Su(H) protein is well visible for one to two hours and decreases conspicuously at the four hours’ time point (Figure 1). Yet, both Su(H)wt-myc and Su(H)LLL-myc protein variants resisted the degradation along the antero-posterior boundary, i.e. in the domain of proteasomal dysfunction (Figure 2b,c), indicating that either protein isoform is subjected to proteasomal degradation.
- We fully agree with the reviewer that fluorescence might be a precarious method for quantifications. Albeit it appears well suitable as signal intensity directly reflects the number of labelled molecules, there are several pitfalls we are well aware off, including antibody batch-variations, the duration of the staining and mounting procedure, the time of storage before acquisition of the pictures, as well as bleaching during the latter process. Also critical are the thickness of the specimen and the exact area where pictures are taken. We took great care to account for all these pitfalls, and proceeded the experiments always in parallel with identical settings. In addition, we used Western blots for protein quantification, holding their own pitfalls, as we well realize and account for to our best knowledge. The overall concordance of the half-life measurements gives us confidence in our data.
Reviewer 2 Report
The manuscript sheds light on the novel role of Suppressor of Hairless Su(H) stability and in turn on the regulation of Notch signaling. The authors have shown that Hairless (H) binding defective Su(H)LL has a shorter half-life compared to its wild-type counterpart. They attributed this change to proteasomal-mediated degradation of Su(H). The authors have also shown in their study that Su(H) levels depend on the presence of Hairless and that Hairless might protect Su(H) from proteasomal-mediated degradation. However, this protection is indistinguishable from both H and NICD. In an interesting set of experiments, the authors have also claimed that endogenous as well as over-expressed (exogenous) Su(H) competes for endogenous H. The research also claims that the stability of Su(H) is regulated by MAPK-dependent phosphorylation.
Overall, this is an interesting story, however, some key points need to be addressed prior to publication.
Major Comments:
1. In figure 2, it’s advisable that the authors should show the quantification of Su(H) in both the A/P domain and the rest of the disc, at different time points.
2. Line 267: The authors have hypothesized that the dose of Hairless is critical for Su(H) stability. It would be good if the authors can show the same at different doses of Hairless by altering the insertion copy number (say 2 doses of UAS-H and vice versa).
3. There are places where authors have over-hypothesized their results. It would be good to put these in the Discussion section. For example, Line 307: “This rather unexpected result might be explained by a proteasomal overload….”
Minor Comments:
1. Line-116 and 114 are duplicates. The authors should rephrase either of these to avoid redundancy.
2. Line 304: it is better to replace “model” with “hypothesis” because I could not see any presented model here to which the authors are referring.
3. Line 312: It is understandable that by “exogenous” the authors are referring to over-expressed Su(H), it would be good to mention it there for ease of readers.
4. It would be good to mention the ICNRAMANK line in the result section where the authors are mentioning it.
5. The mention of the scale bar is missing in the figure legend of Figure 4.
Author Response
We thank the Reviewer for carefully reading the manuscript and helpful insights. The minor points were all edited as suggested. Regarding the major points, we proceeded as follows:
- As suggested by the Reviewer, we now quantified the Su(H)* protein levels inside and outside the ptc-overexpression domain. We added the data in a new Supplemental Figure S1 (Figure S1. Quantification of Su(H) protein amounts upon blocking proteasomal activity).
- The reviewer raises an interesting point: it would be nice to follow the Su(H)-H dose-relationship in a stepwise manner by changing gene doses accordingly. Yet, the respective experiments are rather difficult to perform and would require to rebuilt a number of fly strains or even generate new transgenes. For example, to raise the dose of UAS-H to two, we would need to first recombine the strain with the hs-Su(H)* alleles, and in addition introduce two copies of ptc-Gal4 to account for the UAS-Gal4 dose-relationship. And vice versa. This cannot be achieved in a decent time window, and hence is beyond the scope of this work. It seems, however, worth following up the idea in future experiments.
- We would prefer to leave the explanation in the results section, because it can account for the observed results in Fig. 4 b and 4c. Moreover, we feel that the discussion covers many different aspects already. Raising yet another point seems too confusing.